# Current protected areas provide limited benefits for European river biodiversity

James S. Sinclair [1] ✉, Rachel Stubbington [2], Ellen A. R. Welti [3], Jukka Aroviita[4], Nathan J. Baker [5], Miguel Cañedo-Argüelles[6], Zoltán Csabai [7,8], David Cunillera-Montcusí [9,10], Sami Domisch [11], Martial Ferréol[12], Mathieu Floury[12,13], Marie Anne Eurie Forio[14], Peter L. M. Goethals[14], Alexia M. González-Ferreras [15], Kaisa-Leena Huttunen[16,17], Richard K. Johnson [18], Lenka Kuglerová[19], Aitor Larrañaga [20], Timo Muotka[17], Riku Paavola [21,22], Petr Pařil [23], Jes J. Rasmussen [24], Ralf B. Schäfer [25], Rudy Vannevel[26], Gábor Várbíró [27], Martin Wilkes[28] & Peter Haase [1,29]

Protected areas are a principal conservation tool for addressing biodiversity loss. Such protection is especially needed in freshwaters, given their greater biodiversity losses compared to terrestrial and marine ecosystems. However, broad-scale evaluations of protected area effectiveness for freshwater biodiversity are lacking. Here, we provide a continental-scale analysis of the relationship between protected areas and freshwater biodiversity using 1,754 river invertebrate community time series sampled between 1986 and 2022 across ten European countries. Protected areas primarily benefited poor-quality communities (indicative of higher human impacts) that were protected, or that gained protection, across a substantial proportion of their upstream catchment. Protection had little to no influence on moderate- and high-quality communities, although high-quality communities potentially provide less scope for effect. Our results reveal the overall limited effectiveness of current protected areas for freshwater biodiversity, likely because they are typically designed and managed to achieve terrestrial conservation goals. Broadly improving effectiveness for freshwater biodiversity requires catchment-scale management approaches involving larger and more continuous upstream protection, and efforts to address remaining stressors. These approaches would also benefit connected terrestrial and coastal ecosystems, thus generally helping bend the curve of global biodiversity loss.

Biodiversity is in crisis owing to human-induced global change[1–3]. Extensive actions have been implemented to address these losses, including legislation and agreements to expand the cover of protected areas (PAs), such as the EU Habitats Directive (92/43/EEC) and the Kunming-Montreal Global Biodiversity Framework, which sets a target of 30% global PA coverage by 2030[4]. PAs restrict or reduce human activity in designated locations, such as national parks, nature reserves, or marine sanctuaries, with the aim of maintaining and restoring biodiversity. Whether PAs generally achieve this aim remains unclear. Several broad-scale (i.e., global or continental) studies have investigated the effectiveness of terrestrial and marine PAs, providing insights into their potential for reducing biodiversity loss[5–7],

exploitation[8], and habitat loss[9,10]. However, similar broad-scale perspectives are currently lacking for freshwaters. This knowledge gap is particularly concerning given that freshwater ecosystems harbor a disproportionate amount of global biodiversity by area, and this biodiversity is declining faster compared to terrestrial and marine ecosystems[11–13]. Evidence of PA effectiveness for individual freshwater ecosystems, or freshwaters in individual regions, is currently mixed[14], with some PAs generally benefitting freshwater biodiversity[15,16] whereas others exhibit little to no effect[17–19]. This variability highlights the need for research that evaluates the general effectiveness of PAs for freshwater biodiversity at broader spatial scales.

Inland (i.e., non-marine) PAs may broadly fail to protect freshwater biodiversity because their boundaries and management typically prioritize terrestrial habitats and charismatic taxa[20,21], lack explicit goals for freshwaters, and neglect the needs of freshwater taxa[22–24]. For example, most inland PAs are small, with ~85% less than 10 square kilometers[25]. However, many freshwater ecosystems, particularly larger rivers, can extend across tens to hundreds of kilometers with catchments encompassing thousands of square kilometers[26]. Thus, local river communities can be impacted by upstream terrestrial pollutants and other inputs across broad spatial scales[27–29]. Small-scale protection of a river site can therefore be compromised by inputs arriving from upstream, unprotected areas[30,31]. Small PAs may also only succeed at protecting local habitat, while other key upstream and downstream habitats used by mobile freshwater taxa remain unprotected[32–34].

Evaluating the benefits of inland PAs for freshwaters requires appropriate counterfactuals, i.e., unprotected areas, for comparison. Studies often rely on spatial comparisons of protected and unprotected sites[17,19,35,36], but this may produce biased results due to spatial biases in PA placement. For example, PAs tend to be designated in less impacted, forested, higher elevation areas with little human development[37,38], which may already have high and/or stable biodiversity compared to unprotected sites. These biases often cannot be fully controlled, which makes it difficult to distinguish the effects of protection from pre-existing differences between sites[39]. An alternative approach is to incorporate a temporal component into the spatial comparisons, specifically by comparing the rate of biodiversity change between protected and unprotected sites[6]. This method better evaluates PA effectiveness by using earlier years within sites as the baseline, thus helping determine whether establishing or expanding protection affected biodiversity, and whether biodiversity was lost (or gained) at a faster rate in unprotected sites. However, such temporal comparisons are hindered by the scarcity of high-resolution time-series data.

To address the need for broad-scale, temporal evaluations of PA effectiveness for freshwater biodiversity, we examine 1,754 time series of river invertebrate communities collected between 1986 and 2022 across ten European countries (Fig. 1 and Supplementary Table 1). We focus on river invertebrates because they are key components of freshwater biodiversity that provide important ecosystem functions and services[40], and they exhibit consistent compositional responses to human pressures[41]. These taxa are therefore commonly used as bioindicators and have been monitored globally for decades, including in countries across Europe. Consequently, analysis of European, long-term river invertebrate community data can address the need for broad-scale, temporal evaluations of PA effectiveness for freshwater biodiversity.

We first quantify biodiversity change as site-specific temporal changes in invertebrate abundance, taxonomic richness, and ecological quality (a measure of human impacts based on similarity to communities in least-impacted conditions; see "Methods" section). We then determine whether the rate of biodiversity change differs between sites with and without upstream PAs (as in ref. 6 for terrestrial

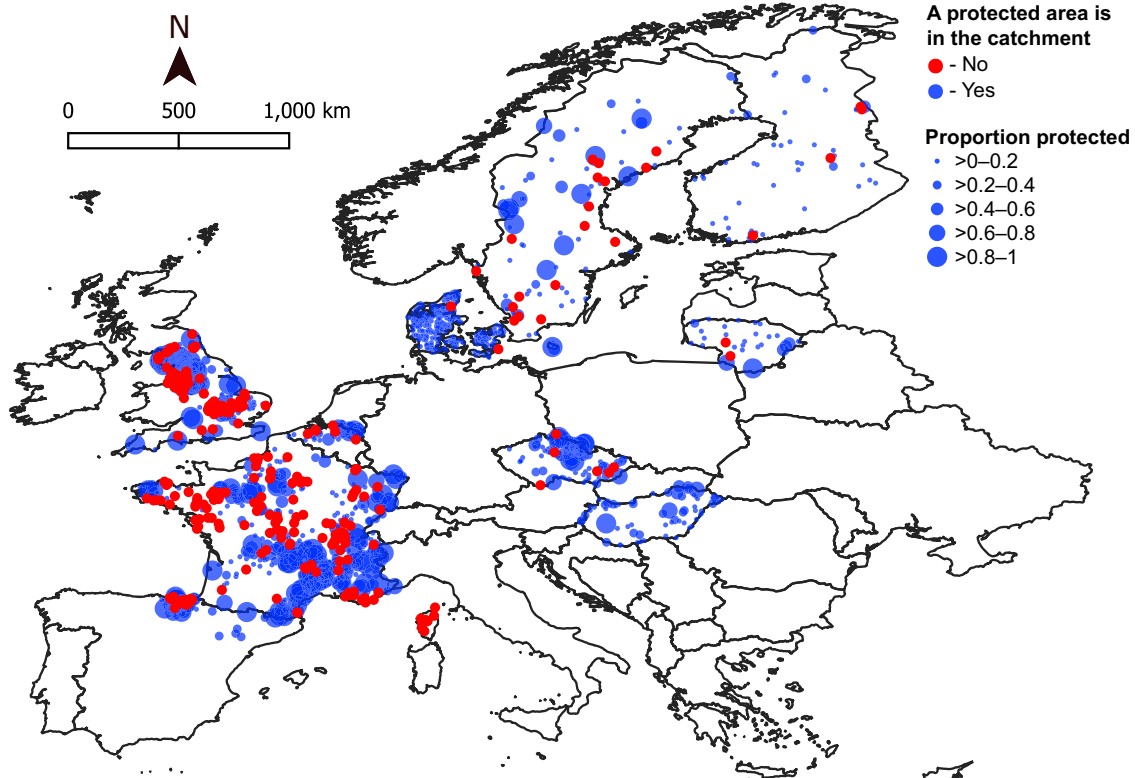

**Fig. 1 | Locations of the 1754 sampled European river sites.** Sites are in Belgium, Czechia, Denmark, Finland, France, Hungary, Lithuania, Spain, Sweden, and the UK. Sites are colored based on the presence of a protected area in the full upstream catchment (no = red; yes = blue). Point sizes for sites with upstream protected areas are based on the proportion of the catchment covered by protected areas.

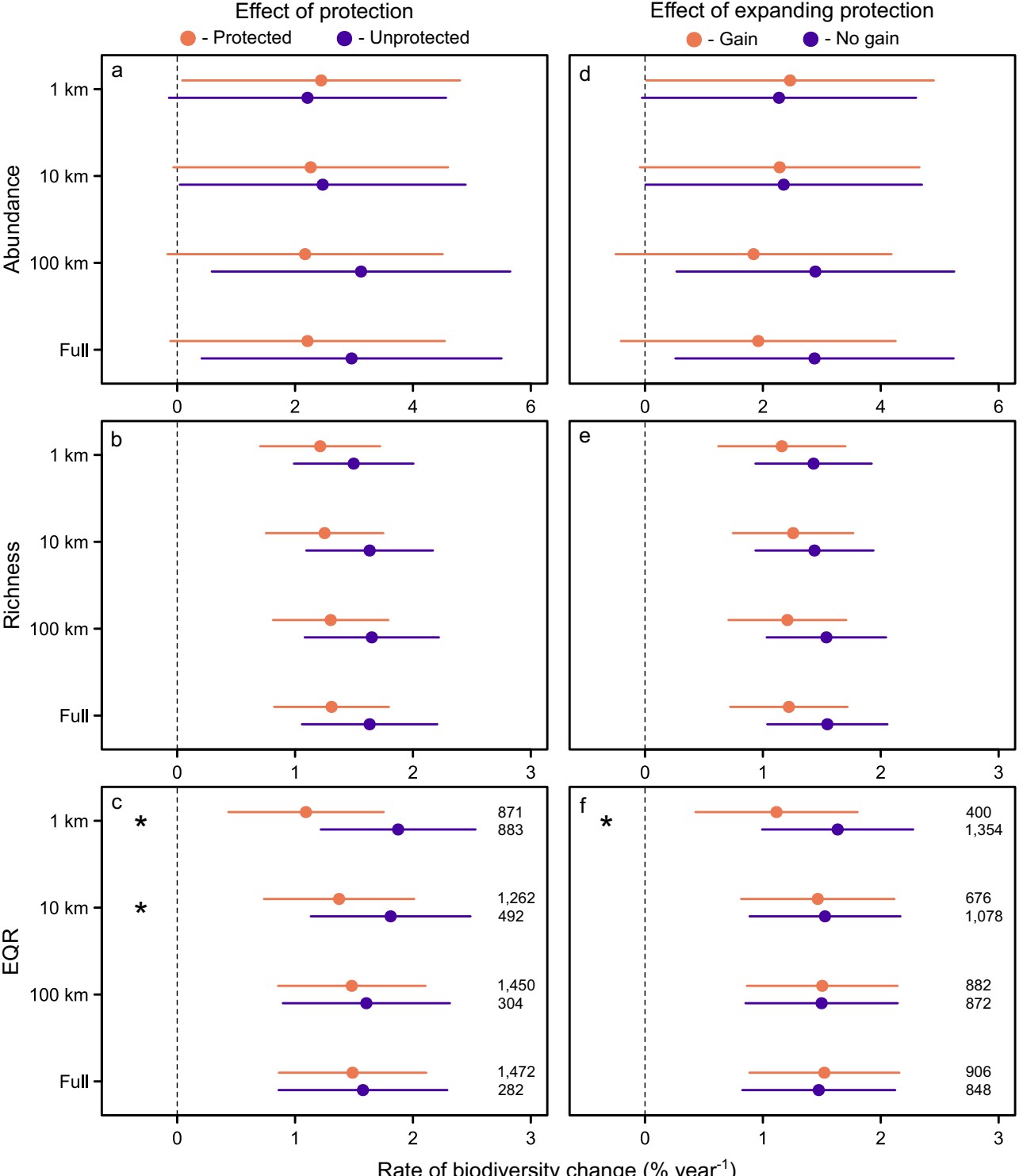

**Fig. 2 | Overall effects of protected areas on river biodiversity.** Rate of temporal change in **a**, **d** abundance, **b**, **e** richness, and **c**, **f** ecological quality (as the Ecological Quality Ratio; EQR) in (**a**–**c**) protected and unprotected sites, and (**d**–**f**) in sites that gained and did not gain upstream PA cover, for the 1-km, 10-km, 100-km, and full upstream scales. Points show the predicted group mean based on the respective linear mixed model, with lines as 95% confidence intervals. Asterisks indicate significant differences between groups based on Likelihood Ratio Tests and corrected for multiple comparisons using the Benjamini–Hochberg false discovery rate (**c**, 1 km: $P < 0.001$, 10 km: $P = 0.021$; **e**, 1 km: $P = 0.039$). Numbers in (**c**, **f**) indicate the number of sites out of 1754 total in each group, and these same sample sizes apply to (**a**, **b**) and (**d**, **e**).

and marine ecosystems), under the expectation that protection would better maintain biodiversity and lead to greater increases in biodiversity through time. To compare the effects of protection close to a river site versus across its broader catchment, we investigate

relationships at four progressively larger upstream distances, ranging from PAs up to 1 km upstream (i.e., the immediate vicinity of a site) 10 km, 100 km, and the full upstream catchment. Lastly, for sites with upstream PAs, we determine whether biodiversity change depends on

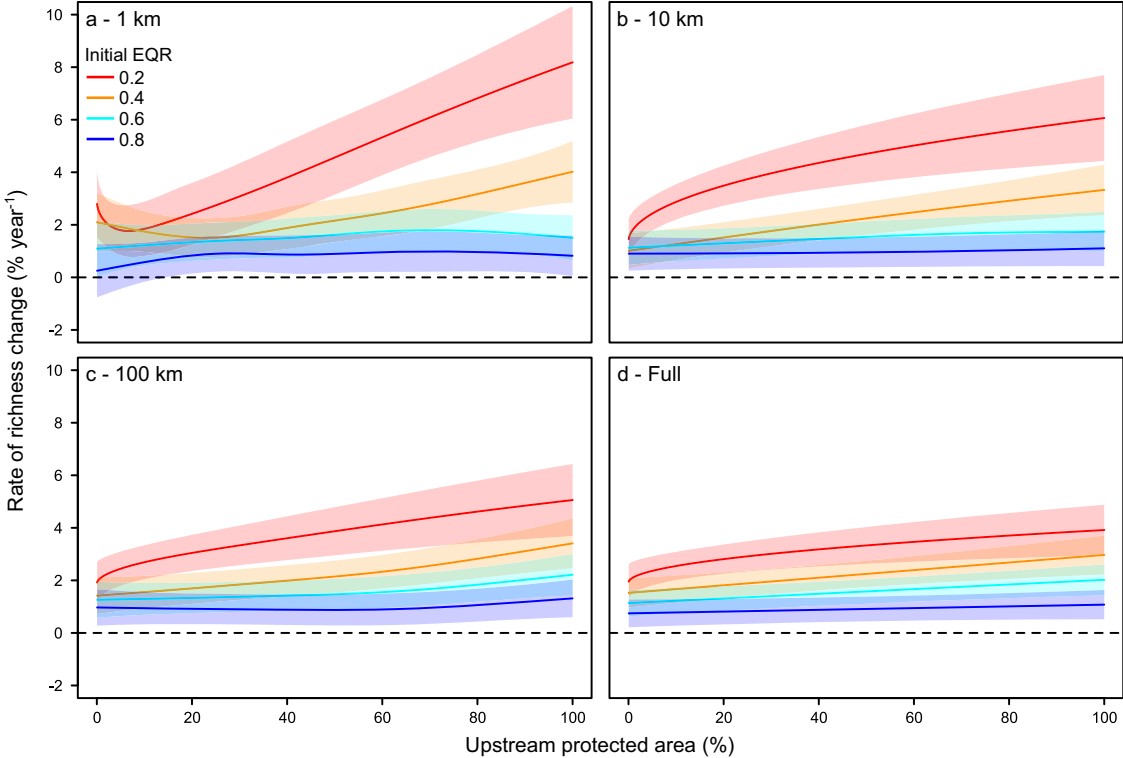

**Fig. 3 | Effects of the amount of protected area on taxon richness.** Relationship between increasing the amount of protected area (PA) cover and the rate of temporal change in richness for only sites with PAs at the **a** 1-km, **b** 10-km, **c** 100-km, and **d** full upstream scales. Lines show the best-fit relationships, with shaded areas as 95% confidence intervals, based on generalized additive mixed models. Line and shading color illustrate how relationships depend on initial ecological quality (as the initial Ecological Quality Ratio, EQR) using examples of 0.2 (red), 0.4 (orange), 0.6 (light blue), and 0.8 (dark blue), which respectively indicate higher to lower human impacts.

the amount of PA cover, or the degree of PA gain, and whether it varies with river size and initial ecological quality. Regarding river size, as discussed above, larger rivers integrate inputs across longer distances, thus potentially exposing their communities to cumulative pollutants from rural and urban sources, so we expect that biodiversity in larger rivers primarily responds to PA cover that spans larger upstream scales. Regarding ecological quality, PAs tend to be designated in already less impacted areas (i.e., better initial ecological quality), and we expect that the effectiveness of such PAs differs from those established in poorer quality sites, which generally have lower biodiversity[42] and thus more scope for improvement.

Here, we show that upstream PAs primarily benefit poor-quality communities where PAs encompass a larger proportion of the catchment. These communities exhibit much higher rates of biodiversity recovery than likely would have occurred in the absence of protection. In contrast, PAs have little to no effect on biodiversity in moderate- and high-quality communities, although the latter group may have been unaffected because human impacts in such rivers are generally low regardless of protection status. Our results underscore the need to broadly improve PA effectiveness in freshwaters by ensuring PA design and management explicitly consider freshwater biodiversity and integrate the needs of freshwater ecosystems.

## Results

### Protected and unprotected sites
Protected and unprotected sites only differed in the rate of ecological quality change (represented as the Ecological Quality Ratio; EQR), and only when protections encompassed smaller upstream scales, specifically when PAs were within 1-km (based on a significant Likelihood ratio test or LRT, $n = 1754$, $L = 23.4$, df = 1, $P < 0.001$) and 10-km upstream distances from a river site (LRT, $n = 1754$, $L = 5.97$, df = 1,

$P = 0.039$; Fig. 2a–c). However, these changes were always greater (i.e., better) in unprotected sites, which was the opposite of our expectations. For example, the rate of EQR change for protected sites at the 1-km upstream scale was +1.1% year$^{-1}$, whereas it was +1.9% year$^{-1}$ for unprotected sites (Fig. 2c). For all other metrics and upstream scales, we found no differences in biodiversity change between protected and unprotected sites (Fig. 2a–c; Supplementary Table 2), with similar proportions of sites in these groups both gaining and losing biodiversity (Supplementary Fig. 1).

For sites with upstream PAs, higher PA cover was related to greater increases in taxon richness and ecological quality, but the nature of these relationships depended on initial ecological quality and upstream scale, as evidenced by significant PA cover*ecological quality interactions from generalized additive mixed models (Supplementary Table 3), and differences in effect sizes among upstream scales. Richness primarily increased with higher PA cover close to a site (i.e., at smaller upstream scales), and primarily in initially degraded communities (i.e., lower initial ecological quality; Fig. 3). For example, considering richness at the 1-km upstream scale and an initially poor ecological quality of 0.2 (i.e., 20% similarity to reference conditions), increasing PA cover from <1% to 100% almost tripled the rate at which richness increased, from +2.8% year$^{-1}$ to +8.2% year$^{-1}$ (Fig. 3a). These effects weakened as the upstream scale and initial ecological quality increased (Fig. 3b–d) to the point that, at the full upstream scale and an initially high ecological quality of 0.8, increasing PA cover from <1% to 100% only increased the rate at which richness increased from +0.7% to +1.1% year$^{-1}$ (Fig. 3d). Similar to richness, ecological quality also increased with higher PA cover primarily in initially poorer quality communities. However, the effect of upstream scale was the opposite to that observed for richness, with greater improvements in ecological quality when PA cover increased at larger upstream scales

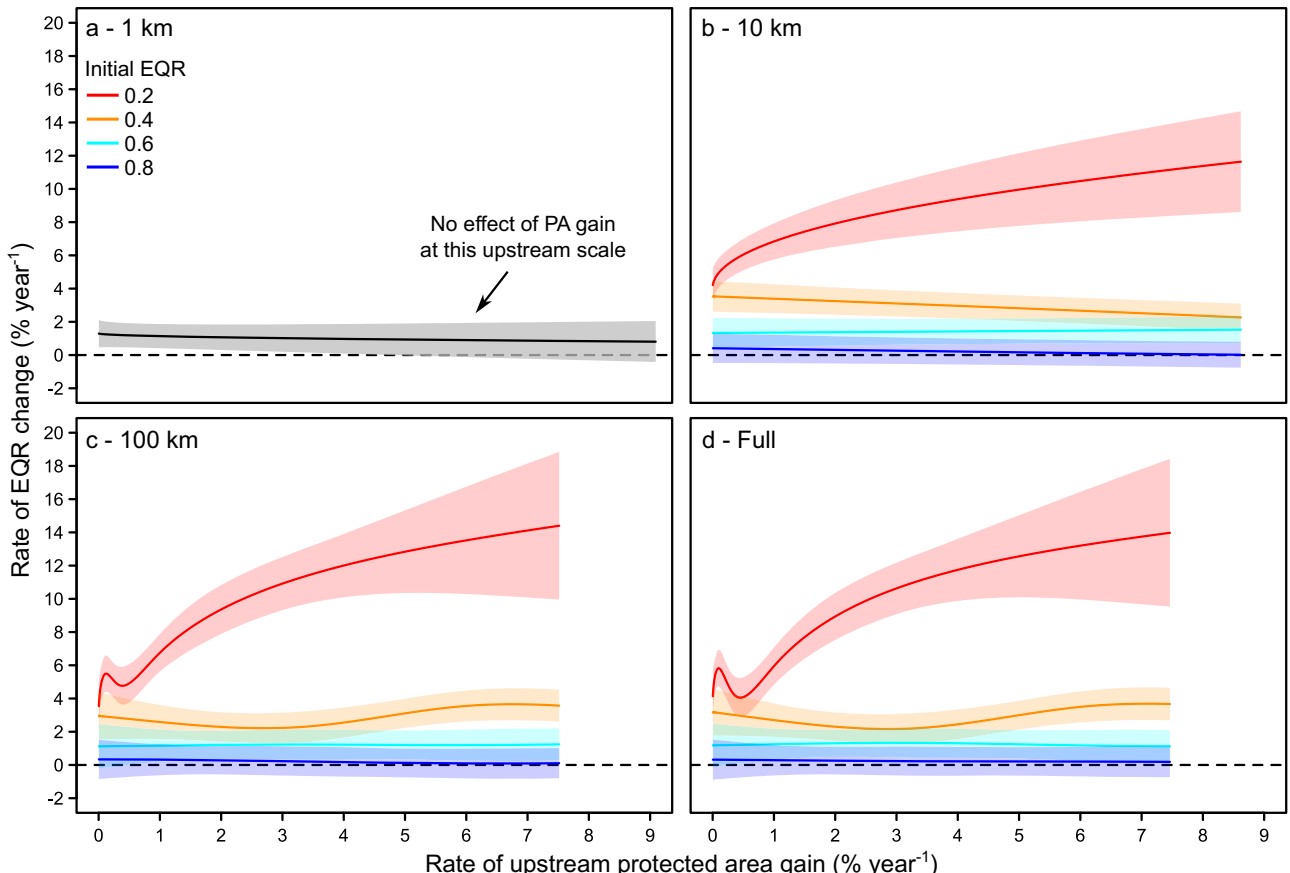

**Fig. 4 | Effects of the rate of protected area gain on ecological quality.** Relationship between gain of upstream protected area (PA) cover and the rate of temporal change in ecological quality (represented as the Ecological Quality Ratio; EQR) for only sites that gained PA cover at the **a** 1-km, **b** 10-km, **c** 100-km, and **d** full upstream scales. Lines show the best-fit relationships, with shaded areas as the 95% confidence intervals, based on generalized additive mixed models. Line and shading color illustrate how relationships depend on initial ecological quality using example initial EQRs of 0.2 (red), 0.4 (orange), 0.6 (light blue), and 0.8 (dark blue), which respectively indicate higher to lower human impacts. Black lines and grey shading indicate non-significant ($P > 0.05$) relationships based on Wald-type tests. Best-fit relationships are shown up to the maximum rate of PA gain observed at each upstream scale.

(Supplementary Fig. 2), indicating ecological quality primarily responded to the amount of protection across the catchment.

Increasing PA cover did not affect the rate of change in abundance at any upstream scale, and we found no evidence for effects of river size (Supplementary Table 3).

## Sites that gained and did not gain protected areas

Similar to the protected and unprotected sites, sites that gained and did not gain PA cover only differed in the rate of ecological quality change, and only at the 1-km upstream scale (LRT, $n = 1754$, $L = 7.79$, df = 1, $P = 0.021$), with greater increases in sites that did not gain PA cover (Fig. 2d–f). For all other metrics and upstream scales, we found no differences in biodiversity change between sites that gained and did not gain PA cover (Fig. 2d–f; Supplementary Table 2), with similar proportions of sites in these groups both gaining and losing biodiversity (Supplementary Fig. 1).

For sites that gained upstream PAs, higher gains translated to greater increases in richness and ecological quality, primarily in initially poorer quality communities and only at larger upstream scales (richness: full only, Supplementary Fig. 3; EQRs: 10 km, 100 km, and full, Fig. 4). These increases were stronger for ecological quality and weaker for richness. For example, at the full upstream scale and an initially poor ecological quality of 0.2, increasing the rate of PA gain from <1% year⁻¹ to the maximum observed value of 7.5% year⁻¹ more than tripled the rate of EQR gain, from +4.1% year⁻¹ to +14% year⁻¹ (Fig. 4d). The rate of richness gain almost doubled under the same

conditions, from +2.3% year⁻¹ to +4.5% year⁻¹ (Supplementary Fig. 3). Furthermore, as initial ecological quality increased, we found some instances where higher PA gains translated to slightly lower rates of increase in both ecological quality and richness. Using the full upstream scale as an example and an initially high ecological quality of 0.8, increasing the rate of PA gain from <1% to 7.5% year⁻¹ decreased the rate of EQR gain from +0.32% to +0.19% year⁻¹ (Fig. 4d).

Increasing PA gain did not affect the rate of change in abundance at any upstream scale, and we found no evidence for effects of river size (Supplementary Table 3).

## Discussion

A principal question for evaluating the effectiveness of protection for nature conservation is to determine what would have happened in its absence[39]. Our results show that, broadly speaking, the same changes in river invertebrate biodiversity occurred regardless of the presence or degree of upstream protection, although PAs improved biodiversity outcomes in a subset of poor-quality communities that had or gained PA cover across a larger proportion of their upstream catchment. Additionally, some rivers lost invertebrate biodiversity during our 1986 to 2022 study period, which occurred in a comparable proportion of protected and unprotected sites. We therefore found no consistent evidence that inland PAs have generally benefited European river invertebrate biodiversity, suggesting that PAs may have not benefited water or habitat quality, given that invertebrates are key indicators of both[43]. These findings provide continental-scale support for similar

results from individual freshwater ecosystems and specific regions for invertebrates[18,31], other taxonomic groups (e.g., fish[17–19]), and water quality[18]. This conclusion should not be misconstrued as suggesting that PAs are ineffective, particularly given that it is based on a subset of total freshwater biodiversity and does not address whether PAs achieved the terrestrial conservation goals they are typically designed and managed for, such as reducing habitat loss[9,10]. We also found that PAs increased the rate of improvement in biodiversity and ecological quality for some river invertebrate communities, and other studies have shown PAs benefiting certain, individual freshwater ecosystems and taxonomic groups[14,44]. Our findings do, however, highlight the need to broadly improve the capacity of inland PAs to support freshwater biodiversity.

Our results for poor-quality communities (e.g., around an initial EQR of 0.2 or 20% similarity to reference conditions) suggest that PAs led to greater increases in biodiversity in these sites than would have occurred without protection. The lesser influence of protection on higher quality communities potentially reflects the already low human impacts in these sites, thus biodiversity remained high and stable regardless of protection status. This explanation fits with our results showing low PA effectiveness in high-quality communities (e.g., initial EQR around 0.8) where biodiversity was likely already high, and may explain why protection was sometimes associated with lower biodiversity gains, which may occur if PAs are placed in areas with a lower scope for improvement (e.g., remote, forested catchments[37,38]). However, it does not explain why PAs were less effective for moderate-quality communities (e.g., initial EQR around 0.4–0.6), which have considerable potential for further improvement. A more likely explanation for these communities is that current approaches to implementing inland PAs, which typically focus on management of terrestrial habitats[23], can address some stressors affecting poor-quality rivers, but not other stressors that may be more relevant in higher quality ecosystems. For example, land-use change and pollution are among the principal stressors driving freshwater biodiversity loss[45]. PAs have some capacity to address these stressors by reducing the human activities that cause them, such as deforestation, urban expansion, intensive agriculture, and tourism[10,46]. Doing so can subsequently improve water and habitat quality in hydrologically connected rivers[47,48]. However, as communities recover, other unaddressed stressors may become more relevant, such as upstream flow alterations or climate change[49], thereby limiting PA effectiveness. Maximizing the benefits of PAs for freshwater biodiversity requires that existing management regimes consider both terrestrial- and freshwater-focused actions[23,50], and set specific goals to address the most important stressors in each freshwater ecosystem. Preventing degradation, including in higher quality rivers, also requires conservation actions beyond establishing PAs, such as better wastewater treatment, habitat restoration, and further improvements to land management practices, including reducing micropollutants[11,51].

PA benefits in initially poor-quality communities varied among upstream scales and community metrics, suggesting that the spatial scale of protection determined which community components were affected. Richness primarily responded to the amount of PA cover close to a site (i.e., at smaller upstream scales), whereas ecological quality primarily responded when protection encompassed and expanded across the broader catchment (i.e., at larger upstream scales). Abundance exhibited no response to protection whatsoever. Increases in richness that neither affect abundance nor substantially alter compositional metrics, such as ecological quality, can occur when only numerically rare species increase[52]. Similarly, compositional changes may not affect richness or abundance if new taxa replace previous taxa[53]. Our results could therefore be explained by protection at smaller scales primarily increasing rare taxa, and protection across larger scales producing stronger compositional recovery by replacing tolerant with sensitive taxa. Increasing rare taxa can provide some

benefits, including potentially diversifying and stabilizing ecosystem functions[54,55], but may represent a less desirable outcome compared to substantially improving invertebrate ecological quality, which is a principal indicator of European river health. Therefore, our results suggest that protecting the broader catchment, or at least a large proportion of the catchment and a river's lateral buffer zones, may elicit greater biodiversity benefits. This conclusion supports the value of catchment-scale rather than local-scale approaches to freshwater protection[22,56], including PAs that are configured to protect and connect key longitudinal (upstream to downstream), lateral (riparian and floodplain), and vertical habitats (surface to groundwater)[14,33,57].

An additional solution to improving PA effectiveness for freshwaters could be to further limit human activities within current PA boundaries, given that many still permit continued human use[58], such as land development and resource extraction. Designating stricter PAs that do not allow such activities may reduce human impacts[46], thus potentially benefiting downstream freshwaters. However, evidence that the strictness of a PA's designation determines its conservation benefits is equivocal[59], including in freshwater ecosystems[16,44]. Stricter protection can also counterintuitively lead to worse conservation outcomes by disenfranchising local communities and promoting illegal use of protected resources[60]. Integrating terrestrial with freshwater approaches to PA design and management may be an alternative approach for improving freshwater conservation outcomes[14,50]. Freshwater-focused PAs (e.g., Ramsar wetlands or river PAs[61]) can be designed based on the distribution of both terrestrial and freshwater biodiversity while accounting for habitat connectivity and downstream impacts[22,30,50]. Effective, adequately funded, and co-produced management is also key to PA effectiveness[14,60]. We therefore advocate that freshwater ecosystems would further benefit from inclusion in PA management priorities that integrate the freshwater needs of local communities and stakeholders.

Inland PAs are increasing globally, supported by the 30% by 2030 coverage target set by the Kunming-Montreal Global Biodiversity Framework[4]. These PAs typically prioritize the needs of terrestrial habitats and taxa, raising questions about their benefits for freshwater biodiversity. Our findings, based on European river invertebrate communities, show that PAs have benefited certain freshwater communities, specifically poor-quality communities where protection encompassed a larger proportion of the upstream catchment. All other communities exhibited more limited (or no) effects of protection, although the lack of effect in high-quality communities may have occurred because these communities are less impacted regardless of whether they are protected or not. Improving overall PA effectiveness, particularly in impacted rivers, requires design and management strategies that explicitly integrate the needs of freshwater ecosystems[14,57], including actions that address multiple stressors and continuous coverage that extends over larger upstream distances and lateral buffer zones. Accordingly, a holistic, catchment-scale framework for managing freshwaters is required[14,22,23,62]. Such a framework would better support freshwater biodiversity, including aquatic invertebrates and the ecosystem functions they provide (e.g., prey, nutrient cycling, and decomposition[40]), and would benefit terrestrial ecosystems via aquatic-terrestrial linkages[63] and marine ecosystems via freshwater-marine linkages[64]. Consequently, improving freshwater protection is a critical issue relevant to all ecosystems and is essential to bend the curve of global biodiversity loss.

## Methods

### River invertebrate biodiversity
We collated river invertebrate time series from ref. 42 and from data provided by European freshwater researchers and managers. We defined the following criteria for data inclusion: (i) time series must span a duration of ≥10 years with ≥7 individual sampling years to enable robust estimation of biodiversity change; (ii) within a time

series, samples in different years must be collected using the same methods and from the same three-month season; (iii) data were available at the community-level with taxa identified to a consistent taxonomic level through time (if inconsistent levels were used then taxa were adjusted to the most temporally consistent level); and (iv) ecological quality values could be calculated for each community following methods compliant with the EU Water Framework Directive (see Supplementary Table 4). These criteria allowed the inclusion of data from ten European countries (Fig. 1). Included data encompassed 1754 sites and 24,245 individual years collected between 1986 and 2022. Included time series spanned a mean total duration of 19.7 ± 5.7 years (mean ± SD) with 13.8 ± 5.5 sampling years (Supplementary Table 1). Taxonomic resolution varied among sites, with 57% (993 sites) identified only to the family level or higher, and 43% (761 sites) identified to a mixed resolution, typically a combination of families, genera, and species, with some classified to intermediate (e.g., subfamily) or higher levels (e.g., Oligochaeta at subclass). These taxonomic differences among sites did not influence our results (see Supplementary Fig. 4). Identifications higher than species level introduce some uncertainty, given that we cannot detect potential species shifts occurring within these groups. However, such identifications still reliably reflect overall community responses to environmental change[65,66] and are common in invertebrate research in which many taxa cannot be reliably identified to the species level.

We quantified biodiversity for each site and year based on three community metrics: (i) abundance (total number of individuals), (ii) richness (total number of taxa), and (iii) ecological quality, quantified as the Ecological Quality Ratio (EQR). Ecological quality is commonly used in Europe as a community-based indicator of human impacts, particularly organic pollutants and general environmental degradation[67]. It reflects the compositional similarity of sensitive and tolerant taxa to expected values from least-impacted reference communities, which are defined based on country-specific criteria (Supplementary Table 4). EQRs are a continuous observed-to-expected ratio of this similarity, ranging from 0 (low similarity indicating high human impacts) to 1 (equal to reference conditions indicating low human impacts), although EQRs for some communities can be above 1, reflecting conditions better than the average reference state. We chose EQRs over other compositional metrics, such as temporal β-diversity, because they provide meaningful information not just about whether communities changed, but also how they changed.

Rates of temporal change in each community metric were quantified for each site by relating site-specific abundance, richness, and EQRs to sampling year using the *gls* function from the *nlme* package in R[68,69], then extracting the slope of this relationship. We included a first-order autoregressive structure in each model to control for temporal autocorrelation between successive years. All slopes were converted to percent change per year by log-transforming all metrics prior to modeling, then exponentiating the slopes, subtracting 1, and multiplying by 100. This transformation ensured all rates of biodiversity change had the same units across sites and metrics.

### Protected areas and upstream scales

We obtained vectorial cartographic polygons for inland PAs from Protected Planet[25]. We excluded 2% of European PAs (accounting for 6% of total cover) for which the year of establishment was unknown. We further excluded all point data due to analytical errors that arise when inferring the dimensions of PAs with unknown boundaries[70]. The majority of point data in our included countries (1171 points out of 1247 total or 94%) were natural monuments in Sweden. These PAs have a registered area of 0 km² because they are individual features, such as a single tree or rock formation, thus contributing marginally to total protected area cover. Of the remaining 76 excluded points, 20 were Ramsar wetlands with a total area of 296 km², and 50 were large biosphere reserves with a total area of 94,188 km². To fill the biosphere

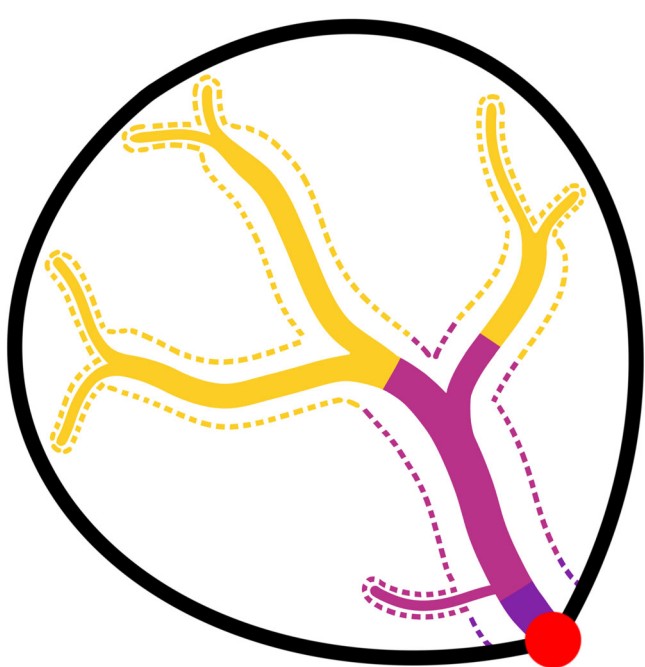

**Fig. 5 | Illustrated example of the four upstream scales.** A river site (red circle) and the upstream scales captured by lateral buffer zones (dashed lines) extending up to 1-km (purple), 10-km (pink), and 100-km (yellow) upstream distances. The solid black outline represents the full upstream contributing area. Note the decline in buffer zone thickness from higher (larger) to lower (smaller) order rivers.

information gap, we used data on European biosphere boundaries from ref. 71, although we excluded the outer transition zones, which are not considered protected. Polygons for all included PAs were dissolved into a single layer, with no distinctions made between different PA types (discussed further in Supplementary Note 1).

In addition to the PA polygons, for each site, we produced upstream polygons representing four different spatial scales. The four scales included: lateral buffer zones extending up the main channel and all tributaries to (i) 1-, (ii) 10-, and (iii) 100-km longitudinal distances; and (iv) the full upstream contributing area, i.e., the upstream catchment including all terrestrial areas that drain into a site. Each upstream scale was selected to account for the potential effects of PAs at progressively greater distances from each site, with the full upstream scale also accounting for PA effects outside the lateral buffer zones. Upstream distances were delineated using the Hydrography90m river network and the *hydrographr* package[72,73]. Buffer zone widths for 1–100-km upstream scales were quantified as 100 m multiplied by the Strahler order of each segment, plus half the predicted river width based on its order from ref. 74 (see Fig. 5). We used this approach because human activities adjacent to a river typically exert the strongest influence on local communities, and 100-m lateral buffer zones effectively capture these effects[75–77]. Additionally, our data encompassed sites from rivers of Strahler orders 1–11, representing small streams to very large rivers, and larger lateral areas are needed to capture the larger surface and ground water inputs to higher-order rivers[78]. Lastly, including river width as part of the buffer zones captured PAs that encompass rivers. The full upstream contributing area was delineated using the *get_upstream_catchment* function from the *hydrographr* package.

Based on the PA and upstream polygons, we calculated the percent of each upstream scale covered by PAs to represent both the presence (>0% cover) and degree (total % cover) of protection. We also calculated the rate of temporal change in percent PA cover to capture biodiversity responses to PA expansion. PA cover was calculated for the year before the first and last year of each invertebrate time series,

which allowed invertebrate communities ≥1 year to respond to environmental changes resulting from PA establishment. Percent PA cover was quantified as the mean percent cover between the first and last years (always ranging from 0 to 100% across sites). Temporal changes in percent PA cover (% year⁻¹) for each site were quantified as the slope of the relationship between PA cover and year, which ranged from 0 to 9% year⁻¹ depending on the upstream scale (1 km: 0–9%; 10 km: 0–8.6%; 100 km: 0–7.5%; Full: 0–7.5% year⁻¹). PA cover changes were only neutral or positive because Protected Planet data cannot represent PA cover declines[79], although PA cover has generally increased globally[58].

In addition to PA cover, we quantified the size (in km²) of each full upstream area to represent river size, given that larger rivers have larger upstream areas. Size was calculated based on the number of 90 m by 90 m pixels in the full upstream area, derived from the Hydrography90m river network. Sites primarily encompassed medium- to larger-sized rivers, with 671 sites out of 1754 total having an upstream catchment size between 10 and 100 km² and 701 sites between 100 and 1000 km², with the remainder comprised of very small (135 sites <10 km²) and very large rivers (247 sites >1000 km²).

## Statistical analyses

We conducted two sets of analyses: (1) categorical comparisons of changes in abundance, richness, and EQRs between protected and unprotected sites (i.e., those with and without upstream PAs), and between sites that did or did not gain upstream PA cover; and (2) for sites with upstream PAs, we related the rate of biodiversity change to the amount of upstream PA cover and the rate of PA gain using regression. The first set of analyses provided a broad overview of the effects of having or gaining any upstream PA cover. The second set determined whether the degree of protection, or its rate of gain, influenced biodiversity change. We also used the second set of analyses to test the influence of river size and initial ecological quality (detailed below). These temporal trend comparisons have some strengths compared to other potential approaches, such as before-and-after comparisons or spatial comparisons of protected and environmentally similar unprotected sites. Specifically, using temporal trends of percent biodiversity change enables comparison of sites that differ in total biodiversity, and allows for variation in protection timing (e.g., sites can be already protected at the start of their time series or can become protected later). Temporal analyses also allow for changes in protection effectiveness through time, such as lagged effects, and capture the potential compounding effects of establishing multiple PAs in subsequent years.

For our first set of analyses, we compared sites with upstream PAs (>0% cover; 'protected') versus those without (0% cover; 'unprotected'), and those that gained upstream PAs (>0% cover year⁻¹; 'gain') versus those that did not (0% cover year⁻¹; 'no gain'). Sites were assigned to these categories separately for each upstream scale, given that sites could change assignments across scales (e.g., an upstream PA is present within 10 km but not 1 km). We then related the continuous, site-level rates of biodiversity change to these categories using linear mixed models (LMMs) conducted in the *lme4* package[80]. Additionally, each model included fixed continuous terms for site latitude and longitude to control for broad-scale spatial trends, a fixed continuous term for time-series length to control for slope differences among shorter to longer time series, and a random intercept term designating the provider of each dataset to control for differences in sampling methods among providers (see Supplementary Table 1). We tested the significance ($P < 0.05$) of the fixed categorical PA term by dropping it from each model and comparing the reduced versus fuller models using Likelihood ratio tests[81]. We ran separate models for each upstream scale. To control for

conducting multiple models using the same response variables, we corrected all $P$-values using the Benjamini–Hochberg false discovery rate[82].

For our second set of analyses, we used generalized additive mixed models (GAMMs) to relate biodiversity change to the amount of upstream PA cover for sites with upstream PAs, and to the rate of PA gain for sites that gained upstream PAs. Models were conducted in the *mgcv* package[83]. PA cover and rates of gain were converted to proportions and square-root transformed prior to analyses to produce a more even distribution of values. To determine the influence of river size, we included an interaction between the PA term and a continuous term for the size of the full upstream area (log₁₀-transformed km²). To determine the influence of initial ecological quality, we included an interaction between the PA term and a continuous term for the EQR averaged across the first three sampling years to represent the initial status of the community. The individual PA, river size, and quality terms were modeled as fixed smoothed terms using thin-plate regression splines, with all fixed interactions modeled using tensor product smooths. Additionally, we included the same fixed and random control variables as for the LMMs. All smoothed terms used the default basis dimensions, and we checked model diagnostics, including the need for higher basis dimensions, using the *gam.check* function. All GAMMs used a Gaussian distribution (identity link function). The significance ($P < 0.05$) of interactions, and if needed the individual PA term, were determined using Wald-type tests[83]. We corrected all $P$-values as above when conducting multiple models using the same response variables.

## Reporting summary

Further information on research design is available in the Nature Portfolio Reporting Summary linked to this article.

## Data availability

All data needed to repeat our analyses are publicly available from https://doi.org/10.6084/m9.figshare.25245430.

## Code availability

All code needed to repeat our analyses is publicly available from https://doi.org/10.6084/m9.figshare.25245430.

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

## Acknowledgements

This study was facilitated by the Senckenberg Synthesis Center on Freshwater Biodiversity Change in Europe and its data providers. We thank Marie-Josée Létourneau for the artwork in Fig. 5 and for permitting its use. J.S.S. and P.H. received funding from the EU Horizon 2020 project eLTER PLUS (#871128). P.H. received funding from the Frontiers Research Foundation (Frontiers Planet Prize). N.J.B. received funding from the Research Council of Lithuania (#S-MIP-24-61) and acknowledges the Lithuanian EPA. D.C.-M. received funding from the EU Horizon 2020 research and innovation programme (#10106238). S.D. received funding from the Leibniz Competition (#J45/2018). A.M.G.-F. acknowledges the grant RYC2023-045780-I funded by MICIU/AEI/10.13039/501100011033 and by ESF+. L.K. received funding from Formas (#2023-00284) and sourced data from the Swedish University of Agricultural Sciences database (Miljödata). A.L. received funding from the Spanish Ministry of Science and Innovation (#PID2020-115830GB-100). P.P. was funded by the Czech Science Foundation (#GA23-05268S), and thanks to CHMI and the Povodí enterprises for the provided data.

## Author contributions

J.S.S. and P.H. conceived the study. J.A., S.D., and R.B.S. contributed to planning and methods. J.S.S. and P.H. wrote most of the manuscript with contributions from all authors, particularly R.S. and E.A.R.W. J.A., N.J.B., M.C.-A., Z.C., D.C.-M,. M.Fe., M.Fl., M.A.E.F., P.L.M.G., A.M.G.-F., K.-L.H., R.K.J., L.K., A.L., T.M., R.P., P.P., J.J.R., R.V., G.V., and M.W. provided invertebrate data or contributed to calculating ecological quality values for their respective countries.

## Funding

## Competing interests

Since April 16th, 2025, Miguel Cañedo-Argüelles has been seconded to the ERC Executive agency. The views expressed in this paper are purely those of the author. They do not necessarily reflect the views or official

positions of the European Commission, the ERC Executive Agency, or the ERC Scientific Council. The other authors declare no competing interests.

## Additional information

[1]Department of River Ecology and Conservation, Senckenberg Research Institute and Natural History Museum Frankfurt, Gelnhausen, Germany. [2]School of Science and Technology, Nottingham Trent University, Nottingham, UK. [3]Division of Biology, Kansas State University, Manhattan, KS, USA. [4]Freshwater and Marine Solutions, Finnish Environment Institute, Oulu, Finland. [5]State Scientific Research Institute Nature Research Centre, Vilnius, Lithuania. [6]SHE2 Research Group, FEHM-Lab (Freshwater Ecology, Hydrology and Management), Institute of Environmental Assessment and Water Research (IDAEA), CSIC, Barcelona, Spain. [7]Department of Hydrobiology, University of Pécs, Pécs, Hungary. [8]HUN-REN Balaton Limnological Research Institute, Tihany, Hungary. [9]Institute of Aquatic Ecology, HUN-REN Centre for Ecological Research, Budapest, Hungary. [10]IFREMER–DYNECO/LEBCO, Centre de Bretagne, Plouzané, France. [11]Leibniz Institute of Freshwater Ecology and Inland Fisheries (IGB), Berlin, Germany. [12]INRAE, UR RiverLy, centre de Lyon-Villeurbanne, Villeurbanne, Cedex, France. [13]University of Paris-Saclay, INRAE, UR HYCAR, Antony, France. [14]Department of Animal Sciences and Aquatic Ecology, Ghent University, Ghent, Belgium. [15]IHCantabria - Instituto de Hidráulica Ambiental de la Universidad de Cantabria, Santander, Spain. [16]Nature Solutions, Finnish Environment Institute, Oulu, Finland. [17]Ecology and Genetics Research Unit, University of Oulu, Oulu, Finland. [18]Department of Aquatic Sciences and Assessment, Swedish University of Agricultural Sciences, Uppsala, Sweden. [19]Department of Forest Ecology and Management, Swedish University of Agricultural Sciences, Umeå, Sweden. [20]Department of Plant Biology and Ecology, University of the Basque Country (UPV/EHU), Leioa, Bilbao, Spain. [21]Oulanka Research Station, University of Oulu Infrastructure Platform, Kuusamo, Finland. [22]Water, Energy and Environmental Engineering Research Unit, Faculty of Technology, University of Oulu, Oulu, Finland. [23]Department of Botany and Zoology, Faculty of Science, Masaryk University, Brno, Czech Republic. [24]Department of Ecoscience, Aarhus University, Aarhus, Denmark. [25]Research Center One Health Ruhr, University Alliance Ruhr & Faculty for Biology, University of Duisburg-Essen, Essen, Germany. [26]Flanders Environment Agency (VMM), Aalst, Belgium. [27]Department of Tisza Research, Institute of Aquatic Ecology, HUN-REN Centre for Ecological Research, Debrecen, Hungary. [28]School of Life Sciences, University of Essex, Colchester, UK. [29]Faculty of Biology, University of Duisburg-Essen, Essen, Germany. ✉e-mail: james.sinclair270@gmail.com

