## [Transparent Peer Review file · Nature Communications]

Current protected areas provide limited benefits for European river biodiversity

Corresponding Author: Dr James Sinclair

Version 0:

Reviewer comments:

Reviewer #1

(Remarks to the Author)

I can see the value of an assessment of this type – particularly building on the uses of long-term data series that have been assembled by this group of authors. The biodiversity trajectory of freshwater ecosystems – and the significant challenges of restoring or protecting what's left – also creates an urgent need to assess the options available. Managing freshwater ecosystems – along with their riparian systems and catchments – is an important option if we can find a way to enforce it while accepting the resource exploitation that would be foregone.

An analysis of the potential collateral downstream benefits from terrestrial protected areas is potentially interesting, but I have some significant questions – and in some cases concerns – about this analysis and write-up as currently presented. Depending on the responses, some analytical refinement may be required.

1) It's germane that terrestrial protected areas (PAs) are seldom notified or designated with the intention of conferring downstream benefits, and even more infrequently with the intention protection of freshwater invertebrates. This is a point for discussion.

2) My largest question is with respect to the ways that protected areas were categorised – and linked to the hypothesis being tested. PAs come in several forms only some of which involve biodiversity conservation. The IUCN offers an outline categorisation – in which the key groups are:

- Ia Strict nature reserve
- Ib Wilderness area
- II National park
- III Natural monument or feature
- IV Habitat/species management area
- V Protected landscape or seascape
- VI Protected areas with sustainable use of natural resources

These categories take many different forms or legal instruments in different European nations.

(See <https://portals.iucn.org/library/sites/library/files/documents/pag-021.pdf>)

Significantly, category V PAs are identified for landscape and/or recreational purposes – often allowing continued resource exploitation such as agriculture, tourism, commercial forestry or extractive industry – and some contain urban land complete with wastewater discharge. These are often large areas – making up a significant proportion of PAs in each nation by land cover. Yet, experience shows they often fail in biodiversity protection (eg <https://www.sciencedirect.com/science/article/pii/S235198942100295X>) and some (many?) would not qualify as a contribution to national 30 x 30 GBF targets. In the UK, which I know best, some of our category V PAs are sources of nutrients or other pollutants to freshwater ecosystems downstream. Although less of a concern, some PAs in the other categories are notified for geological or geomorphological reasons – and again need not emphasise contemporary biodiversity conservation.

The key question is how such category V PAs were treated in your analysis – and whether they would be expected to confer any downstream biodiversity benefits?

3) Related to 2, there is good evidence that many terrestrial PAs have failed to protect designated biodiversity features in the face of external pressures (eg N deposition, climate change...) or because of weak/limited enforcement (see, for eg, Starnes et al. 2021 – the link is above). This raises a point for discussion: why would they be expected to perform any better for systems downstream? A second question may be about whether any downstream biodiversity benefits vary among national administrations with stronger protection?

4) Freshwater ecosystems are sometimes specifically protected in their own right for example through Ramsar (not so often for rivers) or, more likely, through the Natura 2000 network. While very few of the species or features involved are invertebrates, Natura 2000 rivers are often given enhanced protection – for example through tightened discharge consenting or permits. There is a key question about whether such specific protection has brought detectable benefits that might be indicated using freshwater invertebrates. At present, a far smaller proportion of river length is protected in this way than for the proportion of terrestrial landscapes under PAs.

On detail, I offer the following:

Lines 8 and 9: not surprising since protected areas seldom focus on river catchments but...

Lines 9/10/11 ... offers some evidence that enhanced catchment-scale protection can deliver benefits

Lines 12/13/14: these ideas could profitably be expanded in discussion: are OECMs an opportunity or do we need something more enforceable? Note that this also implies reduced exploitation of other catchment resources – for example for food production.

Fig 1: Are there particular reasons for the gaps in the extent of coverage within and among nations? Spain, for example, is represented mostly by only the Basque region. The 'UK' looks to be just England and excludes Wales, Scotland or N. Ireland. France appears to have gaps that correspond to roughly to the Garonne, Charente, Meuse and large parts of the Seine basin. Germany – the country of the lead authors – is missing completely. I think the rationale for site choice needs to be clearer.

Fig 1: protected area in catchment? I wondered if this could be refined to indicate the extent of protection by area or length?

Fig 1: I wondered about any biases in the types of locations that figure in national monitoring networks that might affect the purposes for which the data are used here? These are typically in higher order systems and upstream catchments of a size suited to WFD purposes.

Line 75: aren't larger rivers also at risk of more extensive effects from urban land, industry and intensive agriculture?

Line 77: this might depend on the type of protected area (see point 2, above).

Line 129: are you actually assessing river protection status? I think you're mostly assessing the possible collateral effects of terrestrial protected areas.

Line 138-139: how would this conclusion look if you specifically assessed protection aimed specifically at freshwater locations – for example Natura 2000 rivers? This is an important question because it helps to appraise whether specific measures aimed at freshwaters deliver better than diffuse terrestrial protection

Line 150: this depends on the type of protected area. IUCN Category V areas – often the largest – need not imply any such benefit, and some protected landscapes in this category are in quite modified landscapes

Line 183-184: 'inland waters' are now recognised in the GBF – but this is recent.

Line 204: data are plural

Line 234: exclusions on the basis of PA category might have been interesting (point 2 again). An area of 5km sq could be quite a large proportion of a lower-order catchment.

Line 241: I guess you mean longitudinal distances along the channel? This could easily be read as lateral buffers up to 10 or 100km wide

Line 241: why not calculate the proportional area of catchment protected?

Line 257: degree of protection defined how?

S J Ormerod Cardiff April 28th 2025

(Remarks on code availability)

Reviewer #2

(Remarks to the Author)

This is a really interesting and important MS, protected areas (PA) are an important part of conservation and biodiversity protection yet how well they protect freshwaters is unknown. A major omission. The authors should be congratulated for pulling together a great dataset. The MS is well written. And modern data analysis methods have been used. I do have some question regarding the analysis and interpretation of the data, which may have a large effect on the MS major conclusion. I would want to see either an alternative analysis or an explanation that I am misunderstanding something (which is quite possible) before I would recommend publication.

1. Biodiversity has many meanings, including genetic diversity within a species, the protection of particular (often charismatic) species to ecosystem functions. The current MS is assessing one part of biodiversity, i.e. community level structural measures, e.g. taxa richness. Additionally, patterns in freshwater invertebrate communities do not necessarily match those from other groups of organisms e.g. fish, algae. I have no problem with the MS assessing invertebrate community structure (indeed this can be seen as a strength), but I do think the MS cannot claim to be assessing freshwater biodiversity, unless you first define what you mean by freshwater biodiversity. (In my following points, I will use freshwater biodiversity to mean invertebrate community structure).

2. Follow on from the point above, the methods state that “consistent taxonomic level through time, typically a combination of families, genera, and species”, I want to see more information on the taxonomic resolution of the data assessed. Perhaps some of the lack of benefits from PA areas is because of relatively coarse taxonomy, e.g. if outside protected areas a minority species within a family have persisted despite most species being extirpated, IDing at family (or genus level) would underestimate the benefits of PA. This is particularly so as some families with many species e.g. chironomids are typically not ID to species.

3. Could the design be effectively turned into a BACI (before after control impact), in that you could compare changes at sites before the introduction of protected areas to after the introduction of protected areas relative to sites over similar periods that never had PA (i.e. positive controls) and, if present sites, that had PA (or effectively had PA given they are in locations with minimum human impact) over the entire period (i.e. negative controls)? Such a change may not be practical, but if it were practical, it would improve your design as the BACI design considers changes that are occurring unconnected with PA, e.g. climate change, reductions in pollution.

4. It seems logical that if PA enhance biodiversity once they are enacted, biodiversity gain would be great in PA than outside protected areas. However, PA are often established in areas which are relatively unimpacted by human activities (the MS acknowledges this “PAs tend to be designated in less impacted, forested, higher elevation areas with little human development”). So, how would you expect biodiversity to change in PA once the protection is legally enacted? If prior to their establishment, PA tended to have had less biodiversity loss than outside PA, then would expect less gain in biodiversity from PA, relative to non-PA, as there is less potential for recovery at PA. And this is indeed what you observe Figure 2c. Such results may not necessarily indicate that PA are not effective, rather they may indicate that prior to protection PA had less biodiversity loss than non-PA. Moreover, results in Figure 2 and 3 further support my suggestion in that in protected areas that had high – moderate EQR prior to protection there was no gain in richness or EQR following protection compared to areas with low EQR prior to protection.

5. My point above is that such results do not necessarily indicate the PA are ineffective at protecting freshwater biodiversity, rather that biodiversity protection by PA is dependent on the loss of biodiversity before PA enactment. (If, for example, no biodiversity has been lost, then PA will not result in a gain in biodiversity). So, the major conclusion conveyed by the title “Current protected areas provide limited benefits for European river biodiversity” maybe misleading. Or am I misunderstanding something fundamental?

6. What to do? A BACI design (see above) would go some way to solving this problem. Additionally, for locations that effectively had protection before the enactment of PA (e.g. because they are in remote or at high elevations), might they be regarded as de facto PA in your analysis. Note de facto means ‘in reality or as a matter of fact’.

7. Alternatively comparing biodiversity (not its change per unit time) in areas that become PA to those that never become PA before protected areas are established would also check my suggestion that PA may have lost less biodiversity prior to their enactment.

Some minor points

8. Line 2, I would think that protected areas are to prevent or reduce biodiversity loss not to mitigate biodiversity loss.

9. Line 25, rather than “providing valuable insights into the strengths and weaknesses of current approaches” I would rather see some brief statement as to the effectiveness (or otherwise) of PA for terrestrial and marine systems.

(Remarks on code availability)

I have not examined to code as I suggest changes to the design are needed (or my points are shown to be not valid).

Version 1:

Reviewer comments:

Reviewer #1

(Remarks to the Author)

I apologise for the lateness of my review - reflecting the effects of annual leave and the scientific conference season.

In general, the authors have expended considerable effort in addressing the reviewers' comments, modifying the manuscript accordingly. I don't have any further comments that are sufficiently serious to hinder publication.

I find this an interesting and valuable contribution that should add to debate and prompt action.

S J Ormerod, Cardiff September 8th 2025

(Remarks on code availability)

Reviewer #2

(Remarks to the Author)

The authors have addressed the previous comments I made and I am happy for this MS to be published.

(Remarks on code availability)

Reviewer Comments and Responses

Manuscript ID: NCOMMS-25-22756

MS title: Current protected areas provide limited benefits for European river biodiversity

Bold comments are the reviewer comments with line numbers from the previous version of the manuscript (termed ‘Original’). Author comments are in non-bold text, with line numbers from the revised version of the manuscript (termed ‘Current’). Comments in italics are the relevant sections directly quoted from the manuscript text. We also highlighted all major edits in yellow both here and in the main text to aid with the review process.

Reviewer 1

I can see the value of an assessment of this type – particularly building on the uses of long-term data series that have been assembled by this group of authors. The biodiversity trajectory of freshwater ecosystems – and the significant challenges of restoring or protecting what’s left – also creates an urgent need to assess the options available. Managing freshwater ecosystems – along with their riparian systems and catchments – is an important option if we can find a way to enforce it while accepting the resource exploitation that would be foregone.

An analysis of the potential collateral downstream benefits from terrestrial protected areas is potentially interesting, but I have some significant questions – and in some cases concerns – about this analysis and write-up as currently presented. Depending on the responses, some analytical refinement may be required.

1) It’s germane that terrestrial protected areas (PAs) are seldom notified or designated with the intention of conferring downstream benefits, and even more infrequently with the intention protection of freshwater invertebrates. This is a point for discussion.

We agree it is important to be clear that inland PAs are typically designed and managed to achieve terrestrial conservation goals, which is likely why they have generally little effect on freshwater biodiversity. We now include this point in the Abstract (Current Lines 11–14) to make it clear for all readers that we are providing a freshwater perspective on a typically terrestrial-focused conservation tool. Regarding the Discussion, we now provide more detail on the terrestrial focus of PA design and management (Current Lines 161–166) and we now remind readers of this fact in our concluding paragraph (Current Lines 230–232).

Current Lines 11–14 (added): “*Our results reveal the overall limited effectiveness of current protected areas for freshwater biodiversity, likely because they are typically designed and managed to achieve terrestrial conservation goals.*”

Original Lines 135–138: “*We are not suggesting PAs are generally ineffective, particularly given they can succeed in their goals for terrestrial ecosystems, such as reducing habitat loss^{39,40}, and have been effective for certain, individual freshwater ecosystems and taxonomic groups^{12,41}.*”

Edited to:

Current Lines 161–166: “*This conclusion should not be misconstrued as suggesting that PAs are generally ineffective, particularly given it is based on a subset of total freshwater biodiversity and does not speak to whether PAs achieved the terrestrial conservation goals they are typically designed and managed for, such as reducing habitat loss^{9,10}. We also found that PAs helped some river invertebrate communities, and other studies have shown PAs benefiting certain, individual freshwater ecosystems and taxonomic groups^{14,44}.*”

Original Lines 183–185: “However, freshwaters are typically overlooked in global conservation²², raising questions about whether current PAs are benefiting these ecosystems.”

Edited to:

Current Lines 230–232: “*However, these PAs typically prioritize the needs of terrestrial habitats and taxa, raising questions about their benefits for freshwater biodiversity.*”

2) My largest question is with respect to the ways that protected areas were categorised – and linked to the hypothesis being tested. PAs come in several forms only some of which involve biodiversity conservation. The IUCN offers an outline categorisation – in which the key groups are:

Ia Strict nature reserve

Ib Wilderness area

II National park

III Natural monument or feature

IV Habitat/species management area

V Protected landscape or seascape

VI Protected areas with sustainable use of natural resources

These categories take many different forms or legal instruments in different European nations.

(See <https://portals.iucn.org/library/sites/library/files/documents/pag-021.pdf>)

Significantly, category V PAs are identified for landscape and/or recreational purposes – often allowing continued resource exploitation such as agriculture, tourism, commercial forestry or extractive industry – and some contain urban land complete with wastewater discharge. These are often large areas – making up a significant proportion of PAs in each nation by land cover. Yet, experience shows they often fail in biodiversity protection (eg <https://www.sciencedirect.com/science/article/pii/S235198942100295X>) and some (many?) would not qualify as a contribution to national 30 x 30 GBF targets. In the UK, which I know best, some of our category V PAs are sources of nutrients or other pollutants to freshwater ecosystems downstream. Although less of a concern, some PAs in the other categories are notified for geological or geomorphological reasons – and again need not emphasise contemporary biodiversity conservation.

The key question is how such category V PAs were treated in your analysis – and whether they would be expected to confer any downstream biodiversity benefits?

We thank you for raising this important question. Our initial analyses actually distinguished between different PA categories. We did not specifically isolate category V, given there is not a strong reason to differentiate this category from, for example, categories IV/VI which also allow for more extensive human activities. However, we did separately examine categories of ‘strict’ (I–II), ‘multi-use’ (III–VI), and ‘other’ (those without an IUCN designation), following similar groupings from Elleason et al. 2021 (10.1007/s13280-020-01426-5).

The problem we ran into is that there are comparatively few sites in our dataset with strict PAs. For example, strict PAs alone only occur in 23 of the total 1,472 sites with upstream PAs. Similarly, few sites gain strict PA cover as most gains occurred in multi-use/other PAs. Our initial approach still enabled some general comparisons between biodiversity responses in protected sites with *any* strict upstream PA cover versus protected sites without, for example (following the approach underpinning Fig. 2):

We again found no general differences in biodiversity responses between categories, suggesting little to no influence of the strictness of protection. However, we prefer not to include this analysis in our main text due to some key problems. First, sites with any upstream strict PAs generally also have upstream multi-use/other PAs, so the ‘strict’ versus ‘not strict’ groupings capture the effect of having a strict PA mixed in with other PA types, rather than the effect of

strict protection itself. Second, the low number of sites with any upstream strict PAs (site numbers shown in panels c and f) mean that many comparisons are not robust. This sample size issue is further exacerbated in the GAMMs, which include interactions with ecological quality and catchment area. Consequently, to examine the influence of gaining strict PAs in the GAMMs, sites with strict PAs would require a gradient of strict PA cover, a gradient of ecological quality, and a gradient of catchment area, which generally does not occur across such a small number of sites.

Therefore, what we can say with our data is that the strictness of protection does not appear to affect biodiversity outcomes, although a more robust dataset is needed to really tackle this question. Such a conclusion is also supported by the terrestrial literature and individual examples from freshwater ecosystems. To outline these points for readers, we added a new section to the supplementary that discusses the potential effects of PA strictness and the limitations of our dataset in addressing this question (now Supplementary information 9), and we added text to the Methods that directs readers to this section (Current Lines 305–307). Additionally, we now outline the equivocal evidence from the literature that protection strictness influences conservation outcomes (Current Lines 214–221).

Current Lines 214–221 (added): “An additional solution to improving PA effectiveness for freshwaters could be to further limit human activities within current PA boundaries, given that many still allow for continued human use⁵⁸, such as land development and resource extraction. Designating stricter PAs that do not allow such activities may reduce human impacts⁴⁶, thus potentially benefiting downstream freshwaters. However, evidence that the strictness of a PA’s designation determines its conservation benefits is equivocal⁵⁹, including in freshwater ecosystems^{16,44}. Stricter protection can also counterintuitively lead to worse conservation outcomes by disenfranchising local communities and promoting illegal use of protected resources⁶⁰.”

Current Lines 305–307 (added): “Polygons for all included PAs were dissolved into a single layer, with no distinctions made between different PA types (discussed further in Supplementary Figs. 5 and 6).”

3) Related to 2, there is good evidence that many terrestrial PAs have failed to protect designated biodiversity features in the face of external pressures (eg N deposition, climate change...) or because of weak/limited enforcement (see, for eg, Starnes et al. 2021 – the link is above). This raises a point for discussion: why would they be expected to perform any better for systems downstream? A second question may be about whether any downstream biodiversity benefits vary among national administrations with stronger protection?

We now do a better job in the Introduction of highlighting that terrestrial and marine PAs have a proven potential for reducing biodiversity and habitat loss (Current Lines 26–29), thus clarifying our expectation that inland PAs may generally confer similar benefits to connected freshwater ecosystems. However, we also agree that many PAs fail in this regard due to a combination of factors, such as unaddressed external pressures, a lack of funding, and ineffective management, all of which undoubtedly vary among national administrations. We now underscore the key importance of effective management and funding in the final sentences of the newly added

Discussion paragraph (Current Lines 221–228).

Original Lines 23–25: “Several broad-scale (i.e., global or continental) studies have investigated the effectiveness of terrestrial and marine PAs^{5–8}, providing valuable insights into the strengths and weaknesses of current approaches.”

Edited to:

Current Lines 26–29: “Several broad-scale (i.e., global or continental) studies have investigated the effectiveness of terrestrial and marine PAs, providing insights into their **potential for reducing biodiversity loss^{5–7}, exploitation⁸, and habitat loss^{9,10}.**”

Current Lines 221–228 (added): “**Integrating terrestrial with freshwater approaches to PA design and management may be an alternative approach for improving freshwater conservation outcomes^{14,50}. Freshwater-focused PAs (e.g., Ramsar wetlands or river PAs⁶¹) can be designed based on the distribution of both terrestrial and freshwater biodiversity while accounting for habitat connectivity and downstream impacts^{22,30,50}. Effective, adequately funded, and co-produced management is also key to PA effectiveness^{14,60}. We therefore advocate that freshwater ecosystems would further benefit from inclusion in PA management priorities that integrate the freshwater needs of local communities and stakeholders.**”

4) Freshwater ecosystems are sometimes specifically protected in their own right for example through Ramsar (not so often for rivers) or, more likely, through the Natura 2000 network. While very few of the species or features involved are invertebrates, Natura 2000 rivers are often given enhanced protection – for example through tightened discharge consenting or permits. There is a key question about whether such specific protection has brought detectable benefits that might be indicated using freshwater invertebrates. At present, a far smaller proportion of river length is protected in this way than for the proportion of terrestrial landscapes under PAs.

This is an interesting question, and is again linked to similar points above regarding the need for PA design and management to consider freshwater ecosystems. There are only ~600 Ramsar sites across Europe designated in our download of the Protected Planet database (out of ~60,000 total PAs), and as you suggest not many of these are captured by our river-focused dataset so we cannot speak to their specific effectiveness. As for Natura 2000, the effectiveness of these PAs is generally unknown (see the 2020 EU report on management effectiveness; 10.2800/717133) and there is currently no way for us to identify river-focused Natura 2000 sites based solely on the available information in the Protected Planet database. We could make some selections using expert advice and educated guesses based on PA design (e.g., some are clearly designed to a river shape), but this would require an involved follow-up study beyond our principal objectives.

To address your concern, we incorporated some text into the new paragraph we added to the Discussion to acknowledge the value of freshwater-focused PAs, including Ramsar wetlands and river-focused PAs (which would include Natura 2000 river PAs), and the need for effective PA design and management (edits shown in our response to your point #3).

On detail, I offer the following:

Lines 8 and 9: not surprising since protected areas seldom focus on river catchments but...

Lines 9/10/11 ... offers some evidence that enhanced catchment-scale protection can deliver benefits

Thankfully we found some evidence of benefits, otherwise the entire manuscript would be just doom and gloom!

Lines 12/13/14: these ideas could profitably be expanded in discussion: are OECMs an opportunity or do we need something more enforceable? Note that this also implies reduced exploitation of other catchment resources – for example for food production.

Rather than recommending a particular type of official or unofficial PA, we now better highlight the need for effective design and management regardless of the type of PA being implemented. Our edits first outline the pros and cons of stricter protection (see response to your point #2). We then discuss the need to incorporate local communities and stakeholders in management, which aligns with the purpose of OECMs while acknowledging that management still needs to be effective (see response to your point #3).

Fig 1: Are there particular reasons for the gaps in the extent of coverage within and among nations? Spain, for example, is represented mostly by only the Basque region. The ‘UK’ looks to be just England and excludes Wales, Scotland or N. Ireland. France appears to have gaps that correspond to roughly to the Garonne, Charente, Meuse and large parts of the Seine basin. Germany – the country of the lead authors – is missing completely. I think the rationale for site choice needs to be clearer.

Gaps were determined by data availability and our inclusion criteria. For example, we have little to no suitable data from many countries (e.g., our German data primarily comes from a single river system, the Rhine, and so was excluded). Rivers in many countries also lack suitable time series, such as France and Spain where rivers often only have enough data collected to meet the minimum reporting requirements of the EU WFD, which may mean only a few years of data with samples collected 3–6 years apart in most regions, making them unsuitable for quantifying robust temporal trends. Additionally, our analysis required ecological quality values for any included site, and such data are not available for some countries (e.g., Wales, Ireland, and Scotland). We revised our Methods to clarify these points for readers (Current Lines 251–262).

Original Lines 201–207: “We defined the following inclusion criteria: (i) time series must span a duration of ≥ 10 years with ≥ 7 individual years of data; (ii) within a time series, samples in different years must be collected using the same methods and from the same three-month season; and (iii) data was available at the community-level, and taxa were identified to a consistent taxonomic level through time, typically a combination of families, genera, and species. These criteria resulted in 24,245 individual years of data for 1,754 sites collected between 1986 and 2022 across ten European countries (Fig. 1).”

Edited to:

Current Lines 251–262: “We defined the following criteria for data inclusion: (i) time series must span a duration of ≥ 10 years with ≥ 7 individual sampling years to enable robust estimation of biodiversity change; (ii) within a time series, samples in different years must be collected using the same methods and from the same three-month season; (iii) data were available at the community-level with taxa identified to a consistent taxonomic level through time (if inconsistent levels were used then taxa were adjusted to the most temporally consistent level); and (iv) ecological quality values could be calculated for each community following methods compliant with the EU Water Framework Directive (see Supplementary Table 4). These criteria allowed for the inclusion of data from ten European countries (Fig. 1). Included data encompassed 1,754 sites and 24,245 individual years collected between 1986 and 2022. Included time series spanned a mean total duration of 19.7 ± 5.7 years (mean \pm SD) with an average of 13.8 ± 5.5 sampling years (see Supplementary Table 1).”

Fig 1: protected area in catchment? I wondered if this could be refined to indicate the extent of protection by area or length?

We modified this figure as suggested to have points sized by the proportion of the full upstream catchment overlapped by protected areas.

Fig 1: I wondered about any biases in the types of locations that figure in national monitoring networks that might affect the purposes for which the data are used here? These are typically in higher order systems and upstream catchments of a size suited to WFD purposes.

Our dataset is biased towards medium to larger sized rivers, with 701 sites out of 1,754 total having a full upstream area between 100–1,000 km². However, we do have 135 sites coming from very small rivers (<10 km²), and 671 sites from smaller to medium sized rivers (10–100 km²), so smaller waterbodies are represented. We added some text to the Methods to provide this detail.

Current Lines 337–343 (added): “In addition to PA cover, we quantified the size (in km²) of each full upstream area to represent river size, given that larger rivers have larger upstream areas. Size was calculated based on the number of 90 m by 90 m pixels in the full upstream area, derived from the Hydrography90m river network. Sites primarily encompassed medium to larger sized rivers, with 671 sites out of 1,754 total having an upstream catchment size between 10–100 km² and 701 sites between 100–1,000 km², with the remainder comprised of very small (135 sites <10 km²) and very large rivers (247 sites >1,000 km²).”

Line 75: aren't larger rivers also at risk of more extensive effects from urban land, industry and intensive agriculture?

Agreed. We added some text to this effect in this section.

Original Lines 74–76: “Regarding river size, as discussed above, larger rivers integrate inputs

across longer distances, so we expected that biodiversity in larger rivers would respond to PA cover across larger upstream scales.”

Edited to:

Current Lines 77–80: “Regarding river size, as discussed above, larger rivers integrate inputs across longer distances, thus potentially exposing their communities to cumulative pollutants from rural and urban sources, so we expected that biodiversity in larger rivers would respond to PA cover across larger upstream scales.”

Line 77: this might depend on the type of protected area (see point 2, above).

Our edits detailed in our response to your point #2 above should address this concern.

Line 129: are you actually assessing river protection status? I think you’re mostly assessing the possible collateral effects of terrestrial protected areas.

We re-worded this text to avoid implying these are river-focused protected areas.

Original Lines 128–129: “Our results show that, broadly speaking, the same biodiversity changes occurred regardless of river protection status...”

Edited to:

Current Lines 150–152: “Our results show that, broadly speaking, the same changes in river invertebrate biodiversity occurred regardless of the presence or degree of upstream protection ...”

Line 138-139: how would this conclusion look if you specifically assessed protection aimed specifically at freshwater locations – for example Natura 2000 rivers? This is an important question because it helps to appraise whether specific measures aimed at freshwaters deliver better than diffuse terrestrial protection

Our response above to your point #4 should address this concern. The overall message we now better convey is that it may not be so important what particular type of PA is designated (e.g., category I/II versus V/VI, OECM, etc.) because all types can be effective or ineffective, including freshwater-focused PAs such as Ramsar wetlands or Natura 2000 rivers (see human impacts in Ramsar wetlands discussed in Acreman et al. 2020; 10.1111/conl.12684). What likely matters more is whether freshwater biodiversity is incorporated into PA design and management, and that management is effective and adequately funded.

Line 150: this depends on the type of protected area. IUCN Category V areas – often the largest – need not imply any such benefit, and some protected landscapes in this category are in quite modified landscapes

Our response to your point #2 shows the edits made to outline the mixed evidence for the greater

effectiveness of stricter protection. Category V protected areas have not consistently been found to be any less effective than, for example, category IV/VI or sometimes even categories I/II. Elleason et al. 2021 provides a somewhat recent review of this topic (10.1007/s13280-020-01426-5), with Dudley et al. 2016 (10.3390/land5040038) providing a similar overview specific to category V.

Line 183-184: ‘inland waters’ are now recognised in the GBF – but this is recent.

We edited this section to clarify that it is the protected areas that tend to overlook freshwater ecosystems.

Original Lines 183–185: “*However, freshwaters are typically overlooked in global conservation²², raising questions about whether current PAs are benefiting these ecosystems.*”

Edited to:

Current Lines 230–232: “*However, these PAs typically prioritize the needs of terrestrial habitats and taxa, raising questions about their benefits for freshwater biodiversity.*”

Line 204: data are plural

Fixed from “data is” to “data are”.

Line 234: exclusions on the basis of PA category might have been interesting (point 2 again). An area of 5km sq could be quite a large proportion of a lower-order catchment.

We edited this section to provide further detail on these exclusions (Current Lines 297–304). The 5 km² value was for all natural monuments in the Protected Planet point data, but the vast majority of these do not occur in any countries used in our analyses (e.g., many are in Estonia or Slovakia). Those that do are primarily in Sweden, but all of them are natural monuments with a registered area of 0 km² because they are extremely small, encompassing individual features such as a single tree or rock formation. These protected areas effectively contribute nothing to total coverage, even for very small rivers, and there is no method to include them anyway (e.g., creating a buffer zone based on the PA area) because they have area values of 0.

Original Lines 232–235: “*We excluded point data due to difficulties in delineating their PA boundaries. Most point data represent small natural monuments that contribute little to PA cover (area typically < 5 km²), but some are large biosphere reserves. To fill this information gap...*”

Edited to:

Current Lines 297–304: “*We further excluded all point data due to analytical errors that arise when inferring the dimensions of PAs with unknown boundaries⁷⁰. The majority of point data in our included countries (1,171 points out of 1,247 total or 94%) were natural monuments in Sweden. These PAs have a registered area of 0 km² because they are individual features, such as a single tree or rock formation, thus contributing marginally to total protected area cover. Of the*

remaining 76 excluded points, 20 were Ramsar wetlands with a total area of 296 km², and 50 were large biosphere reserves with a total area of 94,188 km². To fill the biosphere information gap...

Line 241: I guess you mean longitudinal distances along the channel? This could easily be read as lateral buffers up to 10 or 100km wide

We now clarify that these are longitudinal distances.

Line 241: why not calculate the proportional area of catchment protected?

Line 257: degree of protection defined how?

We addressed these two comments by clarifying that we did quantify the proportional (i.e., percent) area of each upstream scale that was protected, including the full upstream catchment, and that this percent cover is the ‘degree of protection’.

Original Lines 239–240: *“For each site, we quantified the proportion of four different upstream scales covered by PAs.”*

The opening sentence of this paragraph may have been confusing because it mentions quantifying PA proportions in a section focused on the delineation of the four upstream scales. We edited this for simplicity by now focusing solely on how the different upstream scales were created:

Current Lines 308–309: *“In addition to the PA polygons, for each site we produced upstream polygons representing four different spatial scales.”*

Original Lines 256–263: *“For each site and upstream scale, we calculated the amount of PA cover to represent whether a site was protected or not and its degree of protection. We also quantified the rate of PA temporal change to capture biodiversity responses to PA expansion. PA cover was calculated for the year before the first and last year of each invertebrate time series, which allowed invertebrate communities ≥ 1 year to respond to environmental changes resulting from PA establishment. The amount of PA cover was quantified as the mean percent cover between the first and last years (always ranging from 0–100% across sites). Temporal changes in PA cover (% year⁻¹) for each site were quantified as the slope of the relationship between PA cover and year...”*

In this section, we edited to clarify how percent PA cover was calculated for each upstream scale, and to define the ‘degree of protection’:

Current Lines 325–333: *“Based on the PA and upstream polygons, we calculated the percent of each upstream scale covered by PAs to represent both the presence (> 0% cover) and degree (total % cover) of protection. We also calculated the rate of temporal change in percent PA cover to capture biodiversity responses to PA expansion. PA cover was calculated for the year before the first and last year of each invertebrate time series, which allowed invertebrate communities \geq*

1 year to respond to environmental changes resulting from PA establishment. Percent PA cover was quantified as the mean percent cover between the first and last years (always ranging from 0–100% across sites). Temporal changes in percent PA cover (% year⁻¹) for each site were quantified as the slope of the relationship between PA cover and year...”

Reviewer 2

This is a really interesting and important MS, protected areas (PA) are an important part of conservation and biodiversity protection yet how well they protect freshwaters is unknown. A major omission. The authors should be congratulated for pulling together a great dataset. The MS is well written. And modern data analysis methods have been used. I do have some question regarding the analysis and interpretation of the data, which may have a large effect on the MS major conclusion. I would want to see either an alternative analysis or an explanation that I am misunderstanding something (which is quite possible) before I would recommend publication.

1. Biodiversity has many meanings, including genetic diversity within a species, the protection of particular (often charismatic) species to ecosystem functions. The current MS is assessing one part of biodiversity, i.e. community level structural measures, e.g. taxa richness. Additionally, patterns in freshwater invertebrate communities do not necessarily match those from other groups of organisms e.g. fish, algae. I have no problem with the MS assessing invertebrate community structure (indeed this can be seen as a strength), but I do think the MS cannot claim to be assessing freshwater biodiversity, unless you first define what you mean by freshwater biodiversity. (In my following points, I will use freshwater biodiversity to mean invertebrate community structure).

Good point! We made some edits to our Discussion to clarify that our results are specific to river invertebrate biodiversity, while also better highlighting that even if our results are limited to this aspect of biodiversity, river invertebrates are key indicators of overall water and habitat quality (Current Lines 150–161).

Original Lines 128–135: “Our results show that, broadly speaking, the same biodiversity changes occurred regardless of river protection status, although PAs improved biodiversity outcomes in a subset of highly impacted rivers that had or gained a higher amount of PA cover. Additionally, some rivers lost invertebrate biodiversity during our 1986–2022 study period, which occurred in a comparable proportion of protected and unprotected sites (see Fig. S1). Current inland PAs have therefore generally not benefited biodiversity in European rivers, providing continental-scale support for similar results reported for individual freshwater ecosystems and regions^{16,17,29}.”

Edited to:

Current Lines 150–161: “Our results show that, broadly speaking, the same changes in river invertebrate biodiversity occurred regardless of the presence or degree of upstream protection, although PAs improved biodiversity outcomes in a subset of poor-quality communities that had or gained PA cover across a larger proportion of their upstream catchment. Additionally, some rivers lost invertebrate biodiversity during our 1986–2022 study period, which occurred in a comparable proportion of protected and unprotected sites. We therefore found no consistent evidence that inland PAs have generally benefited European river invertebrate biodiversity, suggesting that PAs may have also not generally benefited water and habitat quality given that invertebrates are key indicators of both⁴³. These findings provide continental-scale support for

similar results reported from individual freshwater ecosystems and specific regions for invertebrates^{18,31}, other taxonomic groups (e.g., fish¹⁷⁻¹⁹), and water quality¹⁸.”

2. Follow on from the point above, the methods state that “consistent taxonomic level through time, typically a combination of families, genera, and species”, I want to see more information on the taxonomic resolution of the data assessed. Perhaps some of the lack of benefits from PA areas is because of relatively coarse taxonomy, e.g. if outside protected areas a minority species within a family have persisted despite most species being extirpated, IDing at family (or genus level) would underestimate the benefits of PA. This is particularly so as some families with many species e.g. chironomids are typically not ID to species.

We added some detail to our Methods to provide more information on the taxonomic resolution of our data, and we added a supplementary analysis (now Supplementary information 8) to determine whether our results change if we do or do not include sites with only higher-level IDs (i.e., only family level or higher). You are also correct that there could be community changes occurring within higher-level identifications that are not detected by our dataset and we now note this for readers. We further highlight that higher-level identifications still reveal macroinvertebrate community responses to changes in anthropogenic impacts.

Current Lines 262–271 (added): *“Taxonomic resolution varied among sites, with 57% (993 sites) identified only to the family level or higher, and 43% (761 sites) identified to a mixed resolution, typically a combination of families, genera, and species, with some classified to intermediate (e.g., subfamily) or higher levels (e.g., Oligochaeta at subclass). These taxonomic differences among sites did not influence our results (see Supplementary information 9). Identifications higher than species level introduce some uncertainty given we cannot detect potential species shifts occurring within these groups. However, such identifications still reliably reflect overall community responses to environmental change^{65,66} and are common in invertebrate research where many taxa cannot be reliably identified to the species level.”*

3. Could the design be effectively turned into a BACI (before after control impact), in that you could compare changes at sites before the introduction of protected areas to after the introduction of protected areas relative to sites over similar periods that never had PA (i.e. positive controls) and, if present sites, that had PA (or effectively had PA given they are in locations with minimum human impact) over the entire period (i.e. negative controls)? Such a change may not be practical, but if it were practical, it would improve your design as the BACI design considers changes that are occurring unconnected with PA, e.g. climate change, reductions in pollution.

We agree a BACI analysis can be powerful approach, but there are several factors that make our dataset unsuited to this design and better suited for the approach we adopted:

1. The BA component requires a pre/post intervention period, but out of our 1,472 protected sites only 311 (20%) were not protected by the beginning of the collection of each invertebrate time series. Additionally, analyzing this 20% would require splitting each time series into a pre/post period, resulting in many sites having insufficient data for

quantifying robust pre- and post-temporal trends (our requirements are for min. 10 years in duration and min. 7 sampling years).

2. The CI component also requires that the timing of sampling matches between the impacted and control sites, and in our case both would need to come from the same river type and geographic region. In our dataset, there is wide variation both in timing and river characteristics between the protected and unprotected sites, making it infeasible to find appropriate unprotected matches for each of the above 311 protected sites.
3. Furthermore, BACI requires a single intervention, however many of our protected sites experience multiple interventions because multiple PAs are often established in subsequent years. For example, only about 311 sites were unprotected at the start of their time series, but 1,070 sites gain PA cover through time, meaning most of the gains are driven by further PA establishment in already protected sites.

Our approach addresses the above problems because it does not require that each site has a pre/post intervention period, it does not require exactly matching time series for the protected and unprotected sites, and temporal trend analyses can capture potential lagged effects and compounding effects of multiple interventions. To better highlight the strengths of our approach, we added some of the above detail to the section of our Methods that discusses our analyses.

Current Lines 354–361 (added): *“These temporal trend comparisons have some strengths compared to other possible approaches, such as before-after comparisons or spatial comparisons of protected and environmentally similar unprotected sites. First, using temporal trends of percent biodiversity change allows for comparing sites that differ in total biodiversity, and allows for variation in protection timing (e.g., sites can be already protected by the start of their time series or can become protected later). Second, temporal analyses allow for changes in protection effectiveness through time, such as lagged effects, and capture the potential compounding effects of establishing multiple PAs in subsequent years.”*

4. It seems logical that if PA enhance biodiversity once they are enacted, biodiversity gain would be great in PA than outside protected areas. However, PA are often established in areas which are relatively unimpacted by human activities (the MS acknowledges this “PAs tend to be designated in less impacted, forested, higher elevation areas with little human development”). So, how would you expect biodiversity to change in PA once the protection is legally enacted? If prior to their establishment, PA tended to have had less biodiversity loss than outside PA, then would expect less gain in biodiversity from PA, relative to non-PA, as there is less potential for recovery at PA. And this is indeed what you observe Figure 2c. Such results may not necessarily indicate that PA are not effective, rather they may indicate that prior to protection PA had less biodiversity loss than non-PA. Moreover, results in Figure 2 and 3 further support my suggestion in that in protected areas that had high – moderate EQR prior to protection there was no gain in richness or EQR following protection compared to areas with low EQR prior to protection.

You raise a very important point about our need to clarify the conclusions drawn from our results, and what we can and cannot say using our analytical approach. These clarifications are particularly important for ensuring readers do not misconstrue our text as implying that PAs provide no benefits.

Regarding high quality communities (i.e., initial EQR of 0.8), our approach can determine that protection did not alter biodiversity outcomes because protected high-quality sites exhibited the same biodiversity outcomes as unprotected high-quality sites (e.g., the dark blue lines in Figs. 3 & 4). This also means PAs did not maintain biodiversity that would otherwise have been lost without protection. You are, however, correct that we cannot be certain *why* this occurred. It could be that PAs were ineffective, but high-quality communities may also generally experience low human impacts whether they are protected or not, thus providing PAs with a limited scope for effect. The latter explanation is consistent both with why biodiversity change did not differ much between high-quality protected versus unprotected sites, and why biodiversity gain was even sometimes slightly lower with protection.

However, our results and analytical approach still support a conclusion of generally limited PA effectiveness. This is partly based on the various panels of Fig. 2, but also on reduced (or no) PA effectiveness across a variety of moderate-quality communities (e.g., initial EQR of 0.4–0.6 in Fig. 4). These communities have higher initial biodiversity compared to poor-quality communities, but not high enough that there is limited scope for further improvement as they still differ substantially from reference conditions.

To address your concerns, we made a variety of edits throughout our text to clarify our results and conclusions across the gradient of poor to moderate to high quality communities. These edits also somewhat soften the implication that PAs had no effect by more frequently leading with the positive benefits we found in poor-quality communities. The edits were as follows:

Abstract

Original Lines 7–12: *“Contrary to our expectations, site-level rates of biodiversity change were generally unaffected by upstream protection. The only exceptions were some highly impacted sites that had, or gained, protection across a substantial proportion of their catchments, which was associated with a two to three times higher rate of biodiversity gain. To broadly improve the freshwater effectiveness of protected areas...”*

Edited to provide more nuanced results and conclusions across the ecological quality gradient. We also now start this section with the positive results for poor-quality communities, rather than leading with the negative results, placing more emphasis on the benefits PAs provided to certain freshwater ecosystems:

Current Lines 8–14: *“Protected areas primarily benefited poor-quality communities (indicative of higher human impacts) that were protected, or that gained protection, across a substantial proportion of their upstream catchment. Protection had little to no influence on moderate- and high-quality communities, although high-quality communities potentially provide less scope for effect. Our results reveal the overall limited effectiveness of current protected areas for freshwater biodiversity, likely because they are typically designed and managed to achieve terrestrial conservation goals. Broadly improving effectiveness for freshwater biodiversity...”*

Introduction

Nature Communications requires a summary of the results at the end of the Introduction, which has now been added in this revision round. We've taken the opportunity through these additions to further remind readers of how our results varied across the ecological quality gradient, and to lead with the positive results:

Current Lines 84–91 (added): “*Here, we show that upstream PAs primarily benefited poor-quality communities where PAs encompass a larger proportion of the catchment. These communities exhibit much higher rates of biodiversity recovery than likely would have occurred in the absence of protection. In contrast, PAs had little to no effect on biodiversity in moderate- and high-quality communities, although the latter group may have been unaffected because human impacts in such rivers are generally low regardless of protection status. Our results underscore the need to broadly improve PA effectiveness in freshwaters by ensuring PA design and management explicitly consider freshwater biodiversity and integrate the needs of freshwater ecosystems.*”

Results

We added two small sections to the results that specifically detail the effects of protection in high-quality sites (i.e., sometimes slight improvement but sometimes even a reduction in biodiversity gain with protection). These results are then referred to specifically in a modified section of the Discussion (detailed below) that outlines what may have happened in the the high-quality communities:

Original Lines 101–102: “*These effects weakened as the upstream scale and initial ecological quality increased (Fig. 3b–d).*”

Edited to better detail the effects of protection in high quality sites:

Current Lines 114–117: “*These effects weakened as the upstream scale and initial ecological quality increased (Fig. 3b–d) to the point that, at the full upstream scale and an initially high ecological quality of 0.8, increasing PA cover from < 1% to 100% only increased the rate at which richness increased from +0.7% to +1.1% year⁻¹ (Fig. 3d).*”

Current Lines 141–145 (added): “*Furthermore, as initial ecological quality increased, we found some instances where higher PA gains actually translated to slightly lower rates of increase in both ecological quality and richness. Using the full upstream scale as an example and an initially high ecological quality of 0.8, increasing the rate of PA gain from < 1% to 7.5% year⁻¹ decreased the rate of EQR gain from +0.32% to +0.19% year⁻¹ (Fig. 4d).*”

Discussion

We made some modifications to our first Discussion paragraph (Current Lines 161–167) to clarify for readers that we are not suggesting PAs are ineffective, and to better highlight the benefits we found.

Original Lines 135–139: “*We are not suggesting PAs are generally ineffective, particularly given they can succeed in their goals for terrestrial ecosystems, such as reducing habitat loss^{39,40}, and have been effective for certain, individual freshwater ecosystems and taxonomic groups^{12,41}. Our findings do, however, highlight the urgent need to broadly improve the capacity of PAs to support freshwater biodiversity.*”

Edited to:

Current Lines 161–167: “*This conclusion should not be misconstrued as suggesting PAs are generally ineffective, particularly given it is based on a subset of total freshwater biodiversity and does not speak to whether PAs achieved the terrestrial conservation goals they are typically designed and managed for, such as reducing habitat loss^{9,10}. We also found that PAs helped some river invertebrate communities, and other studies have shown PAs benefiting certain, individual freshwater ecosystems and taxonomic groups^{14,44}. Our findings do, however, highlight the need to broadly improve the capacity of inland PAs to support freshwater biodiversity.*”

Additionally, we expanded the section of the Discussion that deals with the reduced effect of protection in higher quality communities. We are now clear that the lack of protection effectiveness in high quality sites could be attributed to their low human impacts and initially high biodiversity, but this explanation likely does not apply to the moderate-quality communities (Current Lines 170–179). Furthermore, we edited our concluding paragraph, which previously focused too strongly on the overall lack of PA effect. We re-worded this section (Current Lines 232–239) to provide a more nuanced overview of our results across the different river types:

Original Lines 142–146: “*That comparable protection did not affect biodiversity in higher quality rivers could reflect their communities’ lower breadth for improvement. However, this explanation is unlikely because protection was less effective even in moderately impacted rivers (e.g., initial EQR around 0.4–0.6) where biodiversity is considerably below reference levels. A more likely explanation is that...*”

Edited to:

Current Lines 170–179: “*The lesser influence of protection on higher quality communities potentially reflects the already low human impacts in these sites, thus biodiversity remained high and stable regardless of protection status. This explanation fits with our results showing low PA effectiveness in high-quality communities (e.g., initial EQR around 0.8) where biodiversity was likely already high, and may explain why protection was sometimes associated with lower biodiversity gains, which may occur if PAs are placed in areas with a lower scope for improvement (e.g., remote, forested catchments^{37,38}). However, it does not explain why PAs exhibited reduced effectiveness in moderate-quality communities (e.g., initial EQR around 0.4–*

0.6), which have considerable potential for further improvement. A more likely explanation for these communities...”

Original Lines 185–188: “Our findings show generally poor effectiveness of PAs for freshwater biodiversity, with benefits limited primarily to highly impacted rivers where PAs encompass a larger proportion of the catchment. Improving overall effectiveness requires PA management strategies to explicitly consider the needs of freshwater ecosystems...”

Edited to:

Current Lines 232–239: “Our findings, based on European river invertebrate communities, show that PAs have benefited certain freshwater communities, specifically poor-quality communities where protection encompassed a larger proportion of the upstream catchment. All other communities exhibited more limited (or no) effects of protection, although the lack of effect in high-quality communities may have occurred because these communities are less impacted regardless of whether they are protected or not. Improving overall PA effectiveness, particularly in impacted rivers, requires design and management strategies that explicitly integrate the needs of freshwater ecosystems...”

5. My point above is that such results do not necessarily indicate the PA are ineffective at protecting freshwater biodiversity, rather that biodiversity protection by PA is dependent on the loss of biodiversity before PA enactment. (If, for example, no biodiversity has been lost, then PA will not result in a gain in biodiversity). So, the major conclusion conveyed by the title “Current protected areas provide limited benefits for European river biodiversity” maybe misleading. Or am I misunderstanding something fundamental?

Our edits to your point #4 above should address this concern by improving the clarity of our conclusions, and demonstrating that our approach can speak to the more nuanced effects of protection across the gradient of poor-, moderate-, and high-quality sites.

6. What to do? A BACI design (see above) would go some way to solving this problem. Additionally, for locations that effectively had protection before the enactment of PA (e.g. because they are in remote or at high elevations), might they be regarded as de facto PA in your analysis. Note de facto means ‘in reality or as a matter of fact’.

7. Alternatively comparing biodiversity (not its change per unit time) in areas that become PA to those that never become PA before protected areas are established would also check my suggestion that PA may have lost less biodiversity prior to their enactment.

We grouped these two points together because one follows from the other. Our above responses outline the unsuitability of our dataset for a BACI-type analysis, and how our approach and results allow us to determine the effects of protection (or lack thereof) across the quality gradient.

Some minor points

8. Line 2, I would think that protected areas are to prevent or reduce biodiversity loss not to mitigate biodiversity loss.

We adjusted this wording to “addressing”.

9. Line 25, rather than “providing valuable insights into the strengths and weaknesses of current approaches” I would rather see some brief statement as to the effectiveness (or otherwise) of PA for terrestrial and marine systems.

We modified the wording to better highlight the positive potential of protected areas in these ecosystems.

Original Lines 23–25: “Several broad-scale (i.e., global or continental) studies have investigated the effectiveness of terrestrial and marine PAs^{5–8}, providing valuable insights into the strengths and weaknesses of current approaches.”

Edited to:

Current Lines 26–29: “Several broad-scale (i.e., global or continental) studies have investigated the effectiveness of terrestrial and marine PAs, providing insights into their potential for reducing biodiversity loss^{5–7}, exploitation⁸, and habitat loss^{9,10}.”